# Effects of Digital Psychotherapy on Suicide: A Systematic Review and Meta-Analysis

**DOI:** 10.3390/healthcare12141435

**Published:** 2024-07-18

**Authors:** Jinseok Oh, Jonggab Ho, Sanghee Lee, Jin-Hyuck Park

**Affiliations:** Special Child Education Research Institute, University of Soonchunhyang, Asan 31538, Republic of Korea; oh486255@naver.com (J.O.); hodori1988@sch.ac.kr (J.H.); sanglh@sch.ac.kr (S.L.)

**Keywords:** suicide, psychotherapy, digital psychotherapy, depression, meta-analysis

## Abstract

Previous studies reported that digital psychotherapy was a clinically beneficial intervention for suicide ideation. However, the effects of digital psychotherapy on other aspects of suicide beyond ideation remain unclear. Therefore, this study investigated the effects of digital psychotherapy on suicide and depression. Articles were identified by searching Cochrane, Google Scholar, Medline, PubMed, Web of Science, and PsycINFO in line with the PRISMA statement, yielding nine randomized controlled trials. The difference between conditions regarding suicide and depression in the effect size of the individual article was calculated using Hedges’ *g*. Most digital psychotherapy interventions were based on cognitive behavioral therapy and delivered via apps or the web for at least six weeks. Suicide outcomes primarily focused on suicide ideation. The findings showed digital psychotherapy achieved a significantly larger effect size for suicide (*g* = 0.488, *p <* 0.001) and depression (*g* = 0.316, *p* < 0.001), compared to controls. Specifically, digital psychotherapy showed a significant effect on both suicide ideation (*g* = 0.478, *p* < 0.001) and other suicidal variables (*g* = 0.330, *p* < 0.001). These results suggest the effectiveness of digital psychotherapy in reducing suicide and depression compared to traditional face-to-face therapy. Future research should consider a wider range of outcomes and examine the long-term effectiveness of digital psychotherapy to better understand its effects on suicide prevention.

## 1. Introduction

Suicide is the most severe consequence of mental health issues, impacting not only individuals but also their families and friends, both directly and indirectly [1]. According to the World Health Organization’s 2019 Suicide Worldwide data, over 700,000 people die by suicide annually, highlighting troubling increases in global suicide rates [2]. These statistics underscore the urgent need for enhanced mental health care to prevent suicide.

Notably, the COVID-19 pandemic has further exacerbated the global demand for mental health care. However, the existing supply of mental health services has struggled to meet this increased demand, prompting many individuals to seek alternative solutions, such as digital mental health services. Advances in information and communication technology have driven the growth of the digital health market, offering new avenues for mental health care [3,4]. Digital mental health services encompass a wide range of offerings, including suicide prevention, mental health promotion, and treatment for drug and alcohol addiction, all delivered through digital platforms such as websites and mobile applications [4].

Digital mental health services offer several advantages: they are not limited by geographic location and can make mental health knowledge more accessible, thereby reducing barriers to care and encouraging wider uptake [4,5]. Consequently, digital psychotherapy has been increasingly applied to various clinical populations with mental health issues. Specifically, digital psychotherapy has proven beneficial for suicide prevention, particularly for individuals with suicidal ideation, for whom face-to-face treatment may pose significant barriers [6].

A prior meta-analysis on digital psychotherapy for suicide prevention included 16 randomized controlled trials to evaluate its efficacy [7]. Among these, 10 studies focused directly on suicide, while six addressed depressive symptoms. The interventions, primarily based on cognitive behavioral therapy or dialectical therapy, were delivered via web or mobile applications. The number of sessions ranged from 4 to more than 10, with the main outcome measures including suicidal ideation and depression. The findings indicated that digital psychotherapy significantly reduced both suicidal ideation and depression compared to waitlist or placebo control groups. Another recent meta-analysis reviewed nine randomized controlled trials of digital psychotherapy for suicide prevention [1]. Of these, three trials were guided by clinical teams and six were self-guided, all utilizing cognitive behavioral therapy. The interventions were delivered via web or mobile applications, with sessions ranging from 2 to 10. The main outcome measures included depression, anxiety, and hopelessness. The findings demonstrated that digital psychotherapy significantly outperformed waitlist or conventional care control groups in reducing depression. Taken together, these studies suggest that digital psychotherapy could be effective in preventing suicide and improving mental health.

However, previous meta-analyses have primarily focused on suicidal ideation and depression, overlooking other critical suicidal variables such as suicide risk, behavior, or severity [1,7]. In the process leading to suicide, not only suicide ideation but also suicide planning and execution are important. Consequently, it is crucial to investigate other suicidal variables alongside suicide ideation. Therefore, this study aimed to conduct a systematic review and meta-analysis of randomized controlled trials to investigate the effects of digital psychotherapy on a broader range of suicidal variables.

## 2. Materials and Methods

This systematic review and meta-analysis followed the Preferred Reporting Items for Systematic Reviews and Meta-Analyses Statement (PRISMA).

### 2.1. Search Strategy

A literature search was completed in April 2024. This search focused on articles published from 2014 to 10 April 2024 to exclude outdated methods. Six databases were searched (Cochrane, Google Scholar, Medline, PubMed, Web of Science, and PsycINFO) in accordance with a previous meta-analysis [7]. The search terms were “suicide” or “suicidal” or “self-injurious behavior” and “psychotherapy” or “therapy” and “web” or “internet” or “online” or “mobile” or ”smartphone” or “phone” or “app” or “mhealth” and “randomise*” or “randomize*”. This study was registered with PROSPERO (ID: CRD42024537058)

### 2.2. Eligibility Criteria

The eligibility criteria for this study were as follows:Study design: randomized controlled trials (RCTs).Population: no restrictions were placed on population.Intervention: (a) interventions related to suicide prevention; (b) interventions that were digitally delivered (web or app); (c) interventions that delivered theory-based therapeutic content (e.g., cognitive behavioral therapy or dialectical behavioral therapy); (d) interventions that were directed toward subjects.Control: controls received treatment-as-usual or minimal attention (e.g., psychoeducation) or were on a waitlist.Outcomes: (a) primary outcomes were pre- and post-test measures of suicidal thoughts and behaviors, and (b) secondary outcomes included the symptoms of depression.Language: studies written in English or Korean.Full-text articles.

### 2.3. Article Selection

The article search and selection processes reviewed the titles and abstracts of the searched articles following a database search. Then two independent authors finalized the article selection based on the eligibility criteria. Disagreements between the authors were resolved through consultation with a third author.

### 2.4. Risk of Bias and Methodological Quality

To investigate the risk of bias in the selected studies, the Risk of Bias Assessment tool for randomized trials with the Review Manager (RevMan) program (version 5.4.1, The Cochrane Collaboration, 2020) was utilized. The risk of bias was determined by selection bias, allocation, detection bias, performance bias, attrition bias, and reporting bias. Three levels of bias (low, unclear, and high) were assigned. The methodological quality of the selected studies was assessed by the PEDro scale. Two authors independently assessed the risk of bias and methodological quality, resolving discrepancies through discussion with a third author.

### 2.5. Data Extraction and Statistical Analysis

Data extraction from the selected studies was performed by two independent authors. Extracted data included: population characteristics, features of digital psychotherapy, control conditions, and primary and secondary outcomes. All data were coded using means, standard deviations, *p*-values, and t-values for both experimental and control groups at pre-test and post-test.

Statistical analysis was conducted using Comprehensive Meta-Analysis 2.0 (Biostat, Englewood, NJ, USA). Heterogeneity was considered acceptable when I^2^ < 50%. For I^2^ values less than 50%, a fixed-effects model was used. Pooled effect sizes were analyzed using Hedges’ g with 95% confidence intervals (CI). Hedges’ *g* adjusts for intervention differences between experimental and control groups (where Hedges’ *g* < 0.3 indicates a small effect, 0.3 ≤ *g* < 0.6 indicates a moderate effect size, and *g* ≥ 0.6 indicates a large effect size). Mean, standard deviation, and sample size were utilized for result calculations and analyses.

The pooled effect sizes and directions of the selected articles were visually represented using a forest plot. Statistical heterogeneity was assessed by I^2^. Egger’s regression test was employed to evaluate publication bias, with a *p*-value above 0.05 indicating no publication bias [8]. Sensitivity analysis, conducted through Hedges’ *g*, verified the robustness of results across varying conditions, excluding studies with outlier results.

## 3. Results

### 3.1. Study Selection

A total of 394 studies were identified in the initial literature review. Among them, 256 duplicate articles were excluded. The titles and abstracts of the remaining 138 articles were reviewed for preliminary screening. Out of these, nine articles that met the inclusion criteria were finally selected (Figure 1).

### 3.2. Characteristics of the Included Studies

A total of 1779 subjects were included (intervention: n = 887, mean group size: n = 98.5, control: n = 892, mean group size: n = 99.1), with ages ranging from 14.8 to 47.46 years (Table 1). The subjects in the included studies were adolescents, adults, or veterans with suicide ideation in the past month. The educational levels of the subjects were not reported uniformly, leading to disparities in reporting.

Most interventions in the study utilized cognitive behavioral therapy (CBT), dialectical behavioral therapy (DBT), or a combination of treatments. Specifically, there were eight CBT-based interventions: Frame-IT program, ibobly program, LEAP, Virtual Hope Box (VHB), LifeApp’tite, Think Life, Online Self-Help for Suicidal Thought, and Living with Deadly Thought (LwDT). Additionally, two interventions were DBT-based: iDBT-ST and LwDT (Living with Deadly Thought) combined with CBT. Regarding delivery methods, the majority of interventions (66.7%) were web-based, while three were app-based. This indicates a preference for web-based programs over app-based ones in the studies.

The intervention periods ranged mostly from at least 6 weeks to 12 weeks, with some studies extending up to 4 months. Various assessment tools were used to evaluate suicide and depression outcomes, including the Beck Scale for Suicide Ideation (BSS), Suicidal Ideation Attributes Scale (SIDAS), Suicidal Ideation Questionnaire (SIQ), Depressive Symptom Inventory-Suicidality Subscale (DIS-SS), Suicide Status Form (SSF), Columbia Suicide Severity Rating Scale (C-SSRS), and Scale for Suicidal Ideation (SSI). The majority of studies focused on assessing suicide ideation, with eight out of nine studies using these measures. Only one study each utilized assessments for evaluating suicide risk (SSF) and suicide severity and suicide behavior (C-SSRS).

For depression assessment, tools such as the Children Depression Rating Scale Revised (CDRS-R), Reynolds Adolescent Depression Scale-2 (RADS-2), Patient Health Questionnaire-9 (PHQ-9), Major Depression Inventory (MDI), and Beck Depression Inventory (BDI) were employed.

The mean PEDro score was 7.1 out of 10, with all nine studies rated low for random sequence generation and incomplete outcome data (Figure 2). However, most of the included studies showed unclear or high risk in blinding of participants and personnel, blinding of outcome assessment, and selective reporting (Figure 2).

### 3.3. Effect Size of Digital Psychotherapy

#### 3.3.1. Effect on Suicide

The analysis revealed significant heterogeneity across the included studies regarding the effect of digital psychotherapy on suicide (I^2^ = 88.73%, *p* < 0.001). Therefore, a random-effects model was used to determine effect sizes. The pooled effect size was found to be moderate and statistically significant (k = 11, *g* = 0.488, 95% CI = 0.224–0.752, *p* < 0.001) when compared to control groups (Table 2). Additionally, Egger’s test indicated no significant publication bias (Egger’s intercept = 1.40, *p* = 0.52).

Sub-group analyses were conducted for each pooled effect size, distinguishing between suicide ideation and other suicidal variables. Significant heterogeneity was observed across the sub-grouped studies (suicide ideation: I^2^ = 88.02%, *p* < 0.001; other suicidal variables: I^2^ = 92.90%, *p* < 0.001). A random-effects model revealed that the pooled effect size on suicide ideation (k = 8, *g* = 0.478, 95% CI = 0.379–0.578, *p* < 0.01) and other suicidal variables (k = 2, *g* = 0.330, 95% CI = 0.182–0.477, *p* < 0.001) was moderate and statistically significant (Figure 3). Additionally, Egger’s test indicated no significant publication bias (suicide ideation: Egger’s intercept = 0.81, *p* = 0.73; other suicidal variables: Egger’s intercept = 6.01, *p* = 0.65).

#### 3.3.2. Effect on Depression

Regarding the effect of digital psychotherapy on depression, the included studies showed no significant heterogeneity (I^2^ = 12.64%, *p* = 0.333). Therefore, a fixed-effect model was utilized. The pooled effect size was moderate and statistically significant (k = 6, *g* = 0.316, 95% CI = 0.207–0.426, *p* < 0.001) compared to control groups (Figure 4, Table 2). Egger’s test also indicated no significant publication bias (Egger’s intercept = −0.06, *p* = 0.94).

## 4. Discussion

### 4.1. Overall Results

This study is a systematic review and meta-analysis investigating the effects of digital psychotherapy on suicide and depression. The findings indicate that digital psychotherapy was more effective than control conditions in preventing suicide and alleviating depression. The effect sizes ranged from small to moderate with statistical significance, which is consistent with findings from previous systematic reviews and meta-analyses [1,18,19].

### 4.2. Characteristics of Digital Psychotherapy

The digital psychotherapy utilized in the included studies primarily employed CBT, a widely recognized intervention for addressing suicide-related behaviors in young individuals. CBT for suicide prevention targets suicidal thoughts and behaviors through several core modules [1,2]. Firstly, it includes strategies for developing skills to identify and distance oneself from thoughts, feelings, and behaviors associated with suicide. Secondly, it focuses on managing emotions and behaviors using relaxation techniques. Thirdly, it emphasizes problem-solving and cognitive restructuring. Previous research has consistently shown that CBT, particularly when tailored for suicide prevention, effectively reduces suicidal thoughts, decreases the frequency of suicide attempts, alleviates symptoms of depression, hopelessness, and anxiety, and enhances problem-solving skills, which aligns with the findings of our study [1,2,3,4]. In addition to CBT, DBT was utilized in four of the included studies. DBT is another well-established intervention for treating adolescent depression and is specifically designed for high-risk groups prone to suicide, characterized by emotional dysregulation and behavioral dysfunction [20]. Studies applying DBT have demonstrated its effectiveness in reducing behaviors associated with emotional dysregulation [20,21], which is consistent with our study’s findings. Previous meta-analyses suggest that DBT may have greater efficacy than CBT in some contexts of suicide prevention. Furthermore, acceptance and commitment therapy, therapeutic evaluative conditioning, and mixed-component approaches have also shown effectiveness in addressing suicide-related issues [18]. In our study, three of the included studies incorporated these alternative approaches alongside CBT, highlighting the need to further explore their efficacy. However, due to the limited number of studies applying these treatments in a manner conducive to comparison with CBT, our analysis did not permit a definitive assessment of their relative effectiveness.

### 4.3. Suicide and Depression Outcomes

To evaluate the effects of digital psychotherapy, the studies included assessments for suicidal variables and depression. Most assessments focused heavily on suicide ideation, indicating that digital psychotherapy primarily targeted reducing suicidal thoughts. Only a few studies included assessments of suicide risk and severity, suggesting that digital psychotherapy addressed more than just ideation. Additionally, we observed differences in how depression assessments were utilized compared to assessments for suicidal variables.

### 4.4. Effectiveness of Digital Psychotherapy

Previous studies have shown that transitioning face-to-face treatment to a digital format positively impacts the reduction of suicidal ideation and depression. as well as lowering suicide risk and improving overall mental health [18]. In another meta-analysis, the effectiveness of digital psychotherapy was assessed by distinguishing between studies that directly targeted suicide prevention and those that indirectly addressed suicide-related factors. The findings indicated that direct interventions for suicide-related issues were more effective in reducing suicidal ideation [7]. Consistent with our findings, previous meta-analyses consistently demonstrate that digital psychotherapy effectively prevents suicide and reduces depression, particularly when directly targeting suicide prevention [7,18].

### 4.5. Comparison with Previous Literature

Unfortunately, previous meta-analyses have only focused on the effect of digital psychotherapy on suicide ideation, without encompassing other aspects of suicide risk or severity. Indeed, most previous studies did not consider suicidal severity, possibly due to the perception that digital psychotherapy can only be applied to individuals with low suicidal severity [21]. In contrast, the significance of this study lies in its analysis of the effects of digital psychotherapy by including not only suicidal ideation but also suicide risk and severity as outcomes. Suicidal ideation is crucially important in predicting suicide. However, in the progression from suicidal ideation to actual suicide planning and execution, there are also various factors related to suicidal risk and severity [10,16,17]. Therefore, to effectively prevent suicide, it is essential to consider not only suicidal ideation but also diverse variables like suicidal risk and severity, comprehensively examining the entire process of suicide. Our study uniquely addresses this gap by holistically analyzing the entire spectrum of suicide, thus providing a more nuanced understanding of the potential of digital psychotherapy in suicide prevention.

### 4.6. Significance of Digital Platform

In this study, digital psychotherapy was delivered via a web- or app-based program, which is more accessible compared to traditional face-to-face approaches. Digital psychotherapy reduces social stigma and has been found effective in treating depression and anxiety in adolescents, offering a cost-effective approach [22]. Furthermore, the internet is easily accessible to individuals experiencing suicidal thoughts and holds the potential to prevent these thoughts from escalating into suicidal behavior or suicide attempts [23]. Therefore, in settings where face-to-face psychotherapy is limited, digital psychotherapy could be an alternative. Specifically, web-based digital psychotherapy is cost-effective and ensures anonymity and confidentiality. Additionally, compared to face-to-face approaches, web-based digital psychotherapy could offer the advantage of providing service by periodically checking client information in addition to real-time services [10,11,14,15,17]. On the other hand, app-based programs follow a similar approach to web-based programs but offer the added benefit of being more accessible on mobile devices, which are easier to carry and use compared to computers [9,12,16]. However, the proportion of web-based digital psychotherapy was higher in the included studies. This suggests that despite the increased number of smartphone users compared to the past, a significant number of people still access the internet via computers. Furthermore, while the differences are not substantial, it is hypothesized that this could be due to the higher development costs associated with app-based digital psychotherapy.

### 4.7. Clinical Implication

In this meta-analysis, we aimed to overcome the limitations of previous meta-analyses by analyzing the effects of suicidal variables without restricting them to suicide ideation. However, most of the included studies primarily assessed suicide ideation, which limits our ability to demonstrate significant differences from prior studies. Nevertheless, our study is significant as it confirmed that digital psychotherapy targeting suicide risk and severity did not differ significantly in content from digital psychotherapy focusing solely on suicide ideation [1,7]. Furthermore, the effect size of digital psychotherapy remained significant even when these additional variables were included. Therefore, this study suggests that effective suicide prevention requires a comprehensive examination of the entire suicide process, considering not only suicide ideation but also various variables such as suicide risk and severity. Consequently, the clinical implication of this study is that digital psychotherapy should be implemented from early interventions aimed at preventing suicide ideation to later interventions designed to mitigate suicide risk and severity, utilizing content based on multiple theories.

### 4.8. Limitation

Although this study analyzed the effects of digital psychotherapy by broadly including suicidal variables, unlike previous meta-analyses, it has several limitations. Firstly, since considerable heterogeneity was observed in the findings related to suicide, its interpretation requires caution. Secondly, while the included studies were selected with careful consideration of various suicidal variables, they did not analyze variables encompassing the entire suicide process, such as suicide plans and attempts. Future studies should expand their scope to include variables that cover the complete trajectory of suicide. Thirdly, digital psychotherapy was proposed as an alternative to face-to-face therapy, but its comparative effectiveness was not conclusively demonstrated. However, given that digital psychotherapy primarily differs in delivery methods, it is anticipated that effectiveness may not significantly differ. Fourthly, this study did not assess the long-term effects of digital psychotherapy. By focusing solely on immediate post-intervention effects, it is limited in its ability to determine the duration of treatment effects. Therefore, future research should adopt a broader perspective by considering a wider range of variables and examining the long-term impacts of both face-to-face and digital psychotherapy to better understand the effects of digital psychotherapy across the entire spectrum of suicide prevention. In addition, since the number of studies on other theories is relatively small compared to CBT, further validation of its effectiveness is necessary in the future.

## 5. Conclusions

This study explored the effects of digital psychotherapy on suicide and depression. The findings demonstrate that digital psychotherapy is more beneficial to prevent suicide and ameliorate depression. These findings suggest that digital psychotherapy could be an alternative option when face-to-face psychotherapy is not available. Future research should consider a broader range of suicide variables and examine the long-term impact of digital psychotherapy to better understand its effects on suicide prevention.

## Figures and Tables

**Figure 1 healthcare-12-01435-f001:**
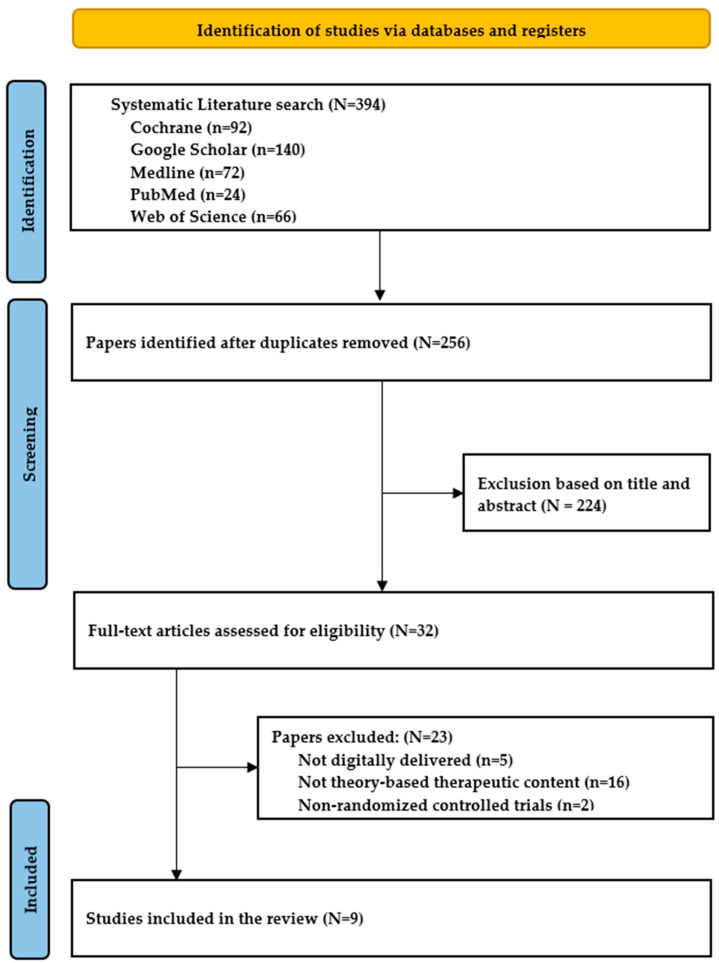
PRISMA Flow chart of the study selection process.

**Figure 2 healthcare-12-01435-f002:**
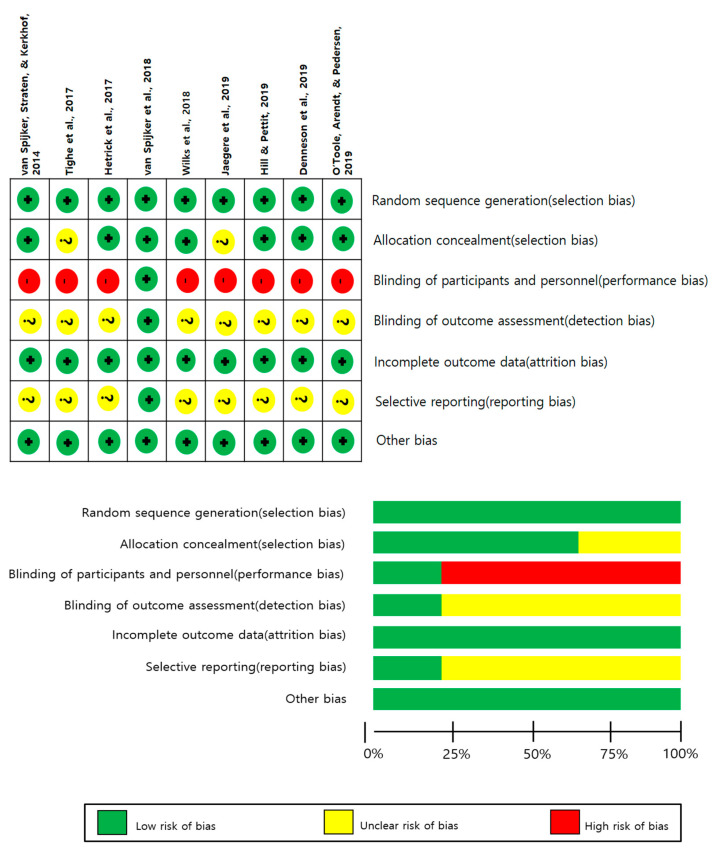
Summary of the risk of bias in the included studies in this meta-analysis. Overall, the risk of bias in the included studies is low. Green with plus mark indicates a low risk of bias, yellow with question mark indicates unclear bias, and red with minus mark indicates a high risk of bias. Van Spijker et al. (2014) [17]; Tighe et al. (2017) [16]; Hetrick et al. (2017) [15]; Van Spijker et al. (2018) [13]; Wilks et al. (2018) [14]; De Jaegere (2019) [10]; Hill et al. (2019) [11]; Denneson et al. (2019) [9]; O’Toole et al. (2019) [12].

**Figure 3 healthcare-12-01435-f003:**
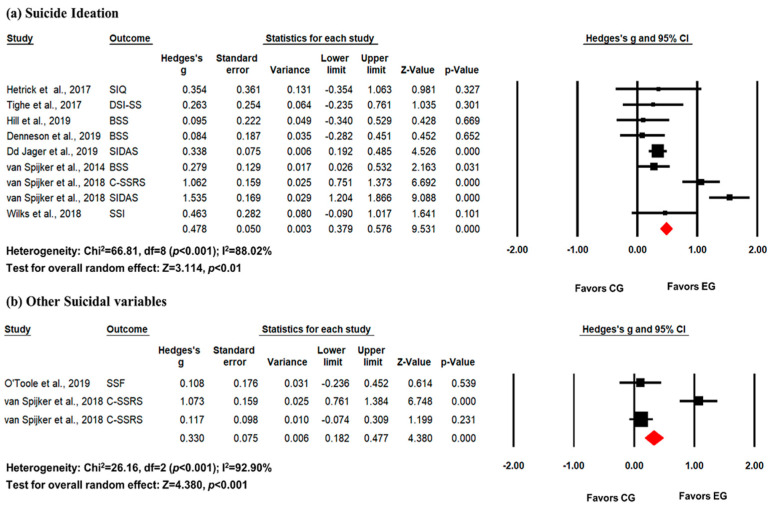
Forest plot for a meta-analysis of the effect of digital psychotherapy on (**a**) suicide and (**b**) other suicidal variables (suicide behavior, risk, and severity). The pooled effect size of digital psychotherapy on suicide and other suicidal variables was moderate and statistically significant. Van Spijker et al. (2018) [13]; Van Spijker et al. (2014) [17]; Tighe et al. (2017) [16]; De Jaegere (2019) [10]; Hetrick et al. (2017) [15]; Hill et al. (2019) [11]; Wilks et al. (2018) [14]; Denneson et al. (2019) [9]; O’Toole et al. (2019) [12].

**Figure 4 healthcare-12-01435-f004:**
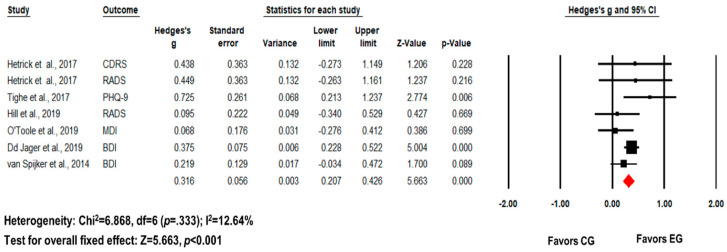
Forest plot for meta-analysis of the effect of digital psychotherapy on depression. The pooled effect size of digital psychotherapy on depression was moderate and statistically significant. Van Spijker et al. (2014) [17]; Tighe et al. (2017) [16]; De Jaegere (2019) [10]; Hetrick et al. (2017) [15]; Hill et al. (2019) [11]; O’Toole et al. (2019) [12].

**Table 1 healthcare-12-01435-t001:** Characteristics of studies included in meta-analysis.

Reference	Study Design	Participants	Intervention	Duration	Outcomes	PEDro-Scale
Denneson et al., 2019 [9]	RCT	Veterans N = 117 (EG = 58, CG = 59)	App-based CBT Virtual Hope Box(VHB)	12 weeks	(Suicide) BSS	7
De Jaegere et al., 2019 [10]	RCT	Adults N = 724 (EG = 365, CG = 359)	Web-based CBT, DBT, and Mindfulness Think Life	12 weeks	(Suicide) BSS/SIDAS (Depression) BDI-II	6
Hill & Pettit 2019 [11]	RCT	Adolescents N = 80 (EG = 31, CG = 30)	Web-based CBT LEAP	6 weeks	(Suicide) BSS (Depression) RADS-2	7
O’Toole, Arendt, and Pedersen, 2019 [12]	RCT	Adults N = 129 (EG = 60, CG = 69)	App-based CBT LifeApp’tite	4 months	(Suicide) SSF (Depression) MDI	7
Van Spijker et al., 2018. [13]	RCT	Adults N = 323 (EG = 160, CG = 163)	Web-based CBT, DBT, and Mindfulness LwDT	6 weeks	(Suicide) C-SSRS/SIDAS (Depression) CES-D	9
Wilks et al., 2018 [14]	RCT	Adults N = 59 (EG = 30, CG = 29)	Web-based DBT Idbt-ST	8 weeks	(Suicide) SSI	7
Hetrick et al., 2017 [15]	RCT	Adolescents N = 50 (EG = 26, CG = 24)	Web-based CBT Reframe-IT	10 weeks	(Suicide) SIQ (Depression) RADS-2/CDRS-R	8
Tighe et al., 2017 [16]	RCT	Youth N = 62 (EG = 31, CG = 30)	App-based CBT ibobbly	6 weeks	(Suicide) DIS-SS (Depression) PHQ-9	6
Van Spijke, Straten, & Kerkhof, 2014 [17]	RCT	Adults N = 236 (EG = 116, CG = 120)	Web-based CBT, DBT, and Mindfulness Online Self-Help for Suicidal Thoughts	18 weeks	(Suicide) BSS (Depression) BDI-II	7

Note: EG = experiment group; CG = control group; CBT = cognitive behavior therapy; DBT = dialectical behavior therapy; BSS = Beck Scale for Suicide Ideation; RADS-2 = Reynolds Adolescent Depression Scale-2; SIDAS = Suicidal Ideation Attributes Scale; BDI-II = Beck Depression Inventory-second edition; SSF = Suicide Status Form; MDI = Major Depression Inventory; C-SSRS = Columbia Suicide Severity Rating Scale; CES-D = Centre for Epidemiological Studies Depression Scale; SSI = Scale for Suicidal Ideation; SIQ = Suicidal Ideation Questionnaire; CDRS-R = Children Depression Rating Scale Revised; DSI-SS = Depressive Symptom Inventory-Suicidality Subscale; PHQ-9 = Patient Health Questionnaire.

**Table 2 healthcare-12-01435-t002:** The summary of the pooled effect size on outcomes.

	Overall Suicide	Suicide Ideation	Other Suicidal Variables	Depression
Hedge’s *g* (95% CI)	0.488 *** (0.224–0.752)	0.478 *** (0.379–0.578)	0.330 *** (0.182–0.477)	0.316 *** (0.207–0.426)
I^2^	88.73%	88.02%	92.90%	12.64%
Egger’s intercept	1.40	0.81	6.01	−0.06

Note: *** *p* < 0.001; CI = confidence interval.

## Data Availability

Data sharing is not applicable; no new data were created or analyzed in this study.

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
