# Peer review of "Effects of Digital Psychotherapy on Suicide: A Systematic Review and Meta-Analysis"

_healthcare, 2024, doi:10.3390/healthcare12141435_

Round 1
Reviewer 1 Report
Comments and Suggestions for Authors
Thank you for this brilliant research. I found it almost ready to be published. I value your strict following PRISMA and transparent description of your study. My only concern is regarding the systematic review. For now, your study looks more like a high-quality meta-analysis instead of a systematic review. I suggest you consider either focusing on your meta-analysis only or re-writing the discussion to make it more appropriate for systematic review. For instance, you could structure the discussion (with headings) and discuss core topics you revealed. After that, the discussion should discuss the topics, which were not covered by the literature you analyzed. So you would show blind spots in this topic. After doing that it could be added to the Abstract. As well as PRISMA could be mentioned in the Abstract too.
Author Response
Reviewer 1
Thank you for this brilliant research. I found it almost ready to be published. I value your strict following PRISMA and transparent description of your study.
My only concern is regarding the systematic review. For now, your study looks more like a high-quality meta-analysis instead of a systematic review. I suggest you consider either focusing on your meta-analysis only or re-writing the discussion to make it more appropriate for systematic review.
For instance, you could structure the discussion (with headings) and discuss core topics you revealed. After that, the discussion should discuss the topics, which were not covered by the literature you analyzed. So you would show blind spots in this topic. After doing that it could be added to the Abstract. As well as PRISMA could be mentioned in the Abstract too.
Response:
Thank you for your constructive comment.
We have re-organized the Discussion section as you suggested. Specifically, we have divided it into 5 sub-headings:
1. Overall Results, 2. Characteristics of Digital Psychotherapy, 3. Suicide and Depression Variables, 4. Effectiveness of Digital Psychotherapy, 5. Comparison with previous literature, 6. Significance of Digital Platform, 7. Clinical Implication, and 8. Limitation.
In addition, the Abstract has been revised as you said.

Reviewer 2 Report
Comments and Suggestions for Authors
Oh et al., discuss the effects of digital psychotherapy on suicide via meta-analysis. The manuscript presents a relative comprehensive investigation into the impact of digital psychotherapy on suicide and depression. In some context, the study is relevant and timely, considering the increasing use of digital health interventions. However, the manuscript requires some revisions before the publication in Healthcare.
1. The novelty and significance of including a broader range of suicide-related variables beyond ideation should be emphasized.
2. The author should mention a more detailed description of the search strategy, including specific keywords used and the rationale for selecting the databases.
3. The inclusion and exclusion criteria are explicitly stated and justified in detail. Especially, there are several errors in Figure.
(1) Papters should be replaced by Papers.
(2) After “Papers identified after duplicates removed”, there are 256 papers remained. Then, 234 papers were removed due to “Exclusion based on title and abstract”. At this time, there should be 22 papers remaining. However, Figure 1 indicated that there were 32 papers remaining. So it is very confusing regarding the exclusion criteria.
4. The results should be clearly presented with appropriate statistical analysis. “This study is a systematic review and meta-analysis to investigate the effects of digital psychotherapy on suicide (g = 0.488, 95% CI = 0.224–0.752, p < 0.001) and depression (g = 0.316, 95% CI = 0.207–0.426, p < 0.001).” in Discussion should be merged into Results.
5. The legends of Figure should be more detailed.
6. Figure 3-5 should be transformed to Tables.
7. The clinical implications of the findings should be discussed in greater detail. For example, How might these results inform the design and implementation of digital mental health interventions?
Comments on the Quality of English LanguageThe quality of English is good.
Author Response
Reviewer 2
Oh et al., discuss the effects of digital psychotherapy on suicide via meta-analysis. The manuscript presents a relative comprehensive investigation into the impact of digital psychotherapy on suicide and depression. In some context, the study is relevant and timely, considering the increasing use of digital health interventions. However, the manuscript requires some revisions before the publication in Healthcare.
Comments 1: The novelty and significance of including a broader range of suicide-related variables beyond ideation should be emphasized.
Response 1: Many thanks for your constructive comment. We have emphasized the significance and novelty of our findings by revising sentences according to your suggestion in the Introduction (“In the process of suicide, not only suicide ideation but also suicide planning and execution are important. Consequently, it is crucial to investigate other suicide-related variables alongside suicide ideation”) and Discussion sections (“Our study uniquely addresses this gap by holistically analyzing the entire spectrum of suicide, thus providing a more nuanced understanding of the potential of digital psychotherapy in suicide prevention”).
Comments 2: The author should mention a more detailed description of the search strategy, including specific keywords used and the rationale for selecting the databases.
Response 2: Many thanks for your keen observation. We have added a more detailed description of the search strategy including keywords and the rationale for selecting the database (“A literature search was completed in April 2024. This search focused on articles published from 2014 to 10 April 2024. Six databases were searched (Cochrane, Google Scholar, MEDLINE, PubMed, Web of Science, PsycINFO) in accordance with a previous meta-analysis [7]. The search terms were “suicide” or “suicidal” or “self-injurious behavior” and “psychotherapy” or “therapy” and “web” or “internet” or “online” or “mobile” or “smartphone” or “phone” or “app” or “mhealth” and “randomis” or “randomize”. This study was registered at the PROSPERO (ID: CRD42024537058)”)
Comments 3: The inclusion and exclusion criteria are explicitly stated and justified in detail. Especially, there are several errors in Figure.
(1) Papters should be replaced by Papers.
Response 3-1: Thank you for your keen observation. Apologies for this typo. We have corrected typo in the Figure 1.
(2) After “Papers identified after duplicates removed”, there are 256 papers remained. Then, 234 papers were removed due to “Exclusion based on title and abstract”. At this time, there should be 22 papers remaining. However, Figure 1 indicated that there were 32 papers remaining. So it is very confusing regarding the exclusion criteria.
Response 3-2: Thank you for your keen observation. Apologies for this confusion. We made typos in Figure 1. We have corrected 234 into 224.
Comments 4: The results should be clearly presented with appropriate statistical analysis. “This study is a systematic review and meta-analysis to investigate the effects of digital psychotherapy on suicide (g = 0.488, 95% CI = 0.224–0.752, p < 0.001) and depression (g = 0.316, 95% CI = 0.207–0.426, p < 0.001).” in Discussion should be merged into Results.
Response 4 : Thank you for your concern. We have removed specific outcome values and statistics in the Discussion. Also, we have revised the Results to clearly present.
Comments 5: The legends of Figure should be more detailed.
Response 5: Thank you for your concern. We have added more detailed information to the captions of the Figures.
Comments 6: Figure 3-5 should be transformed to Tables.
Response 6: Thank you for your supportive comment. We have added the table presenting the findings of the figures.
Comments 7: The clinical implications of the findings should be discussed in greater detail. For example, How might these results inform the design and implementation of digital mental health interventions?
Response 7: Thank you for your constructive comment. Based on the findings of this study, it is recommended that digital psychotherapy should be implemented across various stages: from early interventions aimed at preventing suicidal thoughts to later interventions focused on reducing the severity and risk of suicide. This approach should be grounded in a multi-theory-based framework. We have added this information to the Discussion section (“Consequently, the clinical implication of this study is that digital psychotherapy should be implemented from early interventions aimed at preventing suicide ideation to later interventions designed to mitigate suicide risk and severity, utilizing content based on multiple theories")

Reviewer 3 Report
Comments and Suggestions for Authors
Dear authors, I enjoyed reading your systematic review. Nonetheless, I have a few concerns, but also some suggestions that might help enhance the clarity and comprehensiveness of the research. Below is my point-by-point feedback.
1. Why did your research focus only on articles from 2014 onwards?
2. Which set of keywords did you use?
3. What was Cohen’s kappa concerning author disagreements about study selection, risk of bias, etc.?
4. Why didn’t the review focus solely on CBT? Given that 8 out of 9 studies are about CBT, it seems that your review is more about digital CBT than “general digital therapy”.
5. Maybe I am overlooking something important. What distinguishes the studies included in this review from those in older reviews that focused only on suicidal ideation? It appears that the statistical analyses in your review are based mostly on self-reports assessing suicidal ideation (and depression). Therefore, the statement “previous meta-analyses have primarily focused on suicidal ideation and depression, overlooking other critical suicide-related variables such as suicide risk, behavior, or severity” does not seem supported by the types of articles chosen and the analyses conducted. If your systematic review was solely about the effectiveness of digital therapy on suicidal ideation, how would the study selection and the following statistical analyses differ?
6. A minor thought: Although selecting articles based initially only on titles and abstract is very time-efficient, there is a risk of missing out on useful studies, especially when the focus of the review is very specific. Considering that only a relatively small number of studies (9) were found, a deeper search would have been desirable.
7. In Figure 1, it would be better to specify how many articles were found in each database in the “identification” section.
8. Similar to point 6. In the discussion section, you state “Older studies focused on suicidal ideation, but not on suicide risk and suicide severity.” However, it is hard to grasp the true practical difference. Even if the intents between the reviews (yours included) and the selected studies differ, if, in the end, all reviews select studies about digitally-delivered (CBT) therapies using self-reports such as the Beck Scale for Suicide Ideation (BSS), Suicidal Ideation Attributes Scale (SIDAS), and Suicidal Ideation Questionnaire (SIQ), and conduct meta-analyses on such measures, what is the true difference between the reviews?
9. A minor point: Although only 9 articles were selected, necessitating some compromise, including both web-based and app-based studies (6 and 3, respectively) may significantly increase the heterogeneity of the findings (and maybe significantly skews them too considering the relatively low total number of 9 studies). This consideration should probably be addressed in the limitations section.
Comments on the Quality of English LanguageThe sentence “To investigate the risk of bias in the selected studies, the Risk of Bias Assessment tool for randomized trials with the Review Manager (RevMan) program (version 5.4.1, 104 The Cochrane Collaboration, 2020)” is missing a verb. Please revise for clarity.
Author Response
Reviewer 3
Comments 1: Dear authors, I enjoyed reading your systematic review. Nonetheless, I have a few concerns, but also some suggestions that might help enhance the clarity and comprehensiveness of the research. Below is my point-by-point feedback.
Response 1: Thank you for your supportive comment.
Comments 2: 1. Why did your research focus only on articles from 2014 onwards?
Response 2: Thank you for raising this concern. To exclude old-fashion methods, only papers published in the past 10 years were selected. We have added this information.
Comments 3: 2. Which set of keywords did you use?
Response 3: Many thanks for your keen observation. We have added a more detailed description of the search strategy including keywords and the rationale for selecting the database (“A literature search was completed in April 2024. This search focused on articles published from 2014 to 10 April 2024 to exclude old-fashion methods. Six databases were searched (Cochrane, Google Scholar, MEDLINE, PubMed, Web of Science, PsycINFO) in accordance with a previous meta-analysis [7]. The search terms were “suicide” or “suicidal” or “self-injurious behavior” and “psychotherapy” or “therapy” and “web” or “internet” or “online” or “mobile” or ”smartphone” or “phone” or “app” or “mhealth” and “randomis*” or “randomize*”)
Comment 4: 3. What was Cohen’s kappa concerning author disagreements about study selection, risk of bias, etc.?
Response 4: Thank you for your keen observation. The average Cohen's kappa coefficient across all agreement areas was 0.84. However, according to the standard PRISMA flowchart, this information was not separately included in the main text.
Comments 5: 4. Why didn’t the review focus solely on CBT? Given that 8 out of 9 studies are about CBT, it seems that your review is more about digital CBT than “general digital therapy”.
Response 5: Thank you for your feedback. In this manuscript, we rigorously selected nine papers based on specific review criteria, encompassing Cognitive Behavioral Therapy (CBT), Dialectical Behavior Therapy (DBT), and interventions combining CBT, DBT, and Mindfulness. Among these, eight studies included CBT, with three of them also incorporating DBT. Therefore, it is challenging to attribute the observed effects solely to CBT. However, given the relatively small number of studies utilizing DBT, we acknowledged the reviewer's concerns and highlighted the need for further investigation in the Limitations section of the Discussion (“In addition, since the number of studies on other theories is relatively fewer compared to CBT, further validation of its effectiveness is necessary in the future”)
Comments 6: 5. Maybe I am overlooking something important. What distinguishes the studies included in this review from those in older reviews that focused only on suicidal ideation? It appears that the statistical analyses in your review are based mostly on self-reports assessing suicidal ideation (and depression). Therefore, the statement “previous meta-analyses have primarily focused on suicidal ideation and depression, overlooking other critical suicide-related variables such as suicide risk, behavior, or severity” does not seem supported by the types of articles chosen and the analyses conducted. If your systematic review was solely about the effectiveness of digital therapy on suicidal ideation, how would the study selection and the following statistical analyses differ?
Response 6: Thank you for your constructive comment. Unlike previous studies that focused solely on suicide ideation, this study attempted to encompass a broader range of suicidal variables. While prior studies included only suicide ideation as an outcome, our study, although still primarily focused on suicide ideation, also incorporated measures of suicide behavior, severity (CSSRS), and risk (SSF). This inclusion marks a minimal but notable difference from previous studies.
Nonetheless, we agree with the reviewer. To achieve further differentiation, we have additionally conducted a subgroup analysis instead of sensitivity analysis. Additionally, we have acknowledged as a limitation that most of the results are predominantly based on suicide ideation (“Secondly, while the included studies were selected with careful consideration of suicidal variables, predominantly, suicidal ideation constituted the majority of outcomes, making it difficult to analyze variables encompassing the entire suicide process such as suicide planning and attempts”)
Comments 7: 6. A minor thought: Although selecting articles based initially only on titles and abstract is very time-efficient, there is a risk of missing out on useful studies, especially when the focus of the review is very specific. Considering that only a relatively small number of studies (9) were found, a deeper search would have been desirable.
Response 7: Thank you for your concern. Primary screening based solely on paper titles and abstracts is a common procedure in systematic reviews and meta-analyses. While the concern you mentioned is valid, we believe there are no major issues because the study was conducted independently by two or more researchers and cross-verified for accuracy.
Comments 8: 7. In Figure 1, it would be better to specify how many articles were found in each database in the “identification” section.
Response 8: Thank you for your concern. We have clarified the Figure 1 to provide the information about selection in detail. (“Cochrane: 92, Google Scholar: 140, Medline: 72, PubMed: 24, Web of Science: 66”)
Comments 9: 8. Similar to point 6. In the discussion section, you state “Older studies focused on suicidal ideation, but not on suicide risk and suicide severity.” However, it is hard to grasp the true practical difference. Even if the intents between the reviews (yours included) and the selected studies differ, if, in the end, all reviews select studies about digitally-delivered (CBT) therapies using self-reports such as the Beck Scale for Suicide Ideation (BSS), Suicidal Ideation Attributes Scale (SIDAS), and Suicidal Ideation Questionnaire (SIQ), and conduct meta-analyses on such measures, what is the true difference between the reviews?
Response 9: Thank you for your concern. As mentioned in response to point 5, suicide ideation accounted for most of the outcomes, making it difficult to claim a significant difference between this study and previous studies. However, the significance of this study lies in confirming that the effect size remained significant even when other suicidal variables were included. The addition of a subgroup analysis through this revision appropriately addresses this issue. We hope this clarification is satisfactory.
Comments 10: 9. A minor point: Although only 9 articles were selected, necessitating some compromise, including both web-based and app-based studies (6 and 3, respectively) may significantly increase the heterogeneity of the findings (and maybe significantly skews them too considering the relatively low total number of 9 studies). This consideration should probably be addressed in the limitations section.
Response 10: Thank you for your comment. We have added this limitation to the Discussion section (“Firstly, since considerable heterogeneity was observed in the findings related to suicide, its interpretation with caution is required”)
.
Comments 11: 10. Comments on the Quality of English Language
The sentence “To investigate the risk of bias in the selected studies, the Risk of Bias Assessment tool for randomized trials with the Review Manager (RevMan) program (version 5.4.1, 104 The Cochrane Collaboration, 2020)” is missing a verb. Please revise for clarity
Response 11: Thank you for your keen observation. We have corrected it and this manuscript has been proofread by a native speaker.

Round 2
Reviewer 3 Report
Comments and Suggestions for Authors
The revisions and additions have significantly improved the clarity and overall quality of the paper.